# SARS-CoV-2 Spike-Derived Peptides Presented by HLA Molecules

**Andrea T. Nguyen** [1,2]**, Christopher Szeto** [1,2] **, Dhilshan Jayasinghe** [1,2]**, Christian A. Lobos** [1,2]**, Hanim Halim** [1]**, Demetra S. M. Chatzileontiadou** [1,2]**, Emma J. Grant** [1,2] **and Stephanie Gras** [1,2,*]

[1] Department of Biochemistry and Molecular Biology, Monash University, Clayton, VIC 3800, Australia; A.Nguyen3@latrobe.edu.au (A.T.N.); C.Szeto@latrobe.edu.au (C.S.); D.Jayasinghe@latrobe.edu.au (D.J.); C.Lobos@latrobe.edu.au (C.A.L.); noor.halim@monash.edu (H.H.); D.Chatzileontiadou@latrobe.edu.au (D.S.M.C.); E.Grant@latrobe.edu.au (E.J.G.)

[2] Department of Biochemistry and Genetics, La Trobe Institute for Molecular Science, La Trobe University, Melbourne, VIC 3086, Australia

[*] Correspondence: S.Gras@latrobe.edu.au; Tel.: +61-394-792-569

**Abstract:** The SARS-CoV-2 virus responsible for the COVID-19 pandemic has caused significant morbidity and mortality worldwide. With the remarkable advances in medical research, vaccines were developed to prime the human immune system and decrease disease severity. Despite these achievements, the fundamental basis of immunity to the SARS-CoV-2 virus is still largely undefined. Here, we solved the crystal structure of three spike-derived peptides presented by three different HLA molecules, and determined the stability of the overall peptide–HLA complexes formed. The peptide presentation of spike-derived peptides can influence the way in which CD8[+] T cells can recognise infected cells, clear infection, and therefore, control the outcome of the disease.

**Keywords:** SARS-CoV-2; HLA; peptide presentation; structural biology; spike protein

## 1. Introduction

The SARS-CoV-2 virus is responsible for the ongoing COVID-19 pandemic declared by the World Health Organisation (WHO) in March 2020. After one year of intensive research and clinical trials, some vaccines are currently available and administrated. In addition, we have started to gain an understanding of the immune response towards this emerging virus. However, we still have a lot to discover to understand SARS-CoV-2 infection and the immune response associated with it. Studies reporting some strong T cell and B cell epitopes are emerging [1–4], and this work is paramount to gain an appreciation of the strength and level of immune response that different individuals can produce towards this novel virus.

T cells, and especially CD8[+] or cytotoxic T cells, are critical in the protection against viral infections as they have the capability to recognize and eliminate infected cells in order to clear the infection [5]. CD8[+] T cells recognize peptides derived from the virus that are presented by highly polymorphic human leukocyte antigen (HLA) molecules. In order to understand the CD8[+] T cell response towards the SARS-CoV-2 virus, we need to determine which viral peptides activate T cells, as well as which HLA molecules can stably present them [6,7]. This information, in the context of SARS-CoV-2, is currently limited [8].

Initially, algorithms were used to predict SARS-CoV-2 peptides and their potential HLA restriction. Unfortunately, these predictions are not always accurate and could be attributed to either the wrong HLA molecule or the wrong peptide. Therefore, there is a need to further investigate which peptides are able to bind their predicted HLA molecule, activate a T cell response [9], and thus provide protective immunity. Our previous work showed that not all predicted peptides are able to bind HLA molecules, while others were unstable [7]. Thus, it is critical to have a better understanding of the peptides' ability

to bind and effectively stabilize the peptide–HLA (pHLA) complex, as this impacts how the peptide will be presented, affecting the lifetime that a pHLA can be displayed on the surface of cells, and impacting the likelihood of a T cell binding to the pHLA complex. In addition, structural characterization of peptide presentation by HLA complexes also reveals which peptide residues will be accessible for binding to the T cell receptor (TCR) [6]. This information may help predict and understand which viral mutations within this peptide could be tolerated by T cells or otherwise lead to viral escape, and therefore be of concern [10].

Here, we present the crystal structures of three spike-derived peptides presented by three frequently expressed HLA molecules worldwide, namely HLA-A*02:01, HLA-A*11:01, and HLA-B*35:01. Our work provides a snapshot of the parts of the spike protein that are presented by HLA molecules to the immune system, especially to T cells. In addition, the structures reveal solvent exposed residues in each peptide, which are available for interaction with the TCR. This information could help map potential mutations on these peptides that might be tolerable or detrimental to the immune system.

## 2. Materials and Methods

### 2.1. Protein Expression, Refold, and Purification

DNA plasmids encoding HLA-A*02:01, HLA-A*11:01, and HLA-B*35:01 and human β-2-microglobulin were each separately transformed into a BL21 strain of *Escherichia. coli (E. coli)*. The respective sequences were obtained from the IMGT/HLA database (doi:10.1093/nar/gkz950). The soluble part of the HLA heavy chain (1–275 residues) was ordered sub-cloned into pET30 vector for bacterial expression using the NdeI/HindIII restriction enzyme site for sub-cloning from GenScript. The presence of the insert was confirmed by sequencing by GenScript for each construct. Recombinant proteins were individually expressed and inclusion bodies were extracted and purified from the transformed *E. coli* cells. Thirty milligrams of each of the HLA inclusion bodies was refolded with 10 mg of β-2-microglobulin inclusion bodies and 5 mg of peptide (GenScript, Piscataway, NJ, USA) into a buffer containing 3 M urea, 0.5 M L-arginine, 0.1 M Tris-HCl pH 8.0, 2.5 mM EDTA pH 8.0, 5 mM glutathione (reduced), and 1.25 mM glutathione (oxidised). The peptide sequences are summarized in Table 1. This mixture was dialysed in 10 mM Tris-HCl pH 8.0 and purified using anion exchange chromatography using a Hi-TrapQ column (GE Healthcare, Chicago, IL, USA).

**Table 1.** Stability of pHLA complexes.

| pHLA Complex | Peptide Sequence | Tm [1] ($^\circ$C) |
|---|---|---|
| HLA-A*02:01-S$_{386-395}$ | KLNDLCFTNV | 54.4 $\pm$ 1.3 |
| HLA-A*11:01-S$_{370-378}$ | NSASFSTFK | 54.2 $\pm$ 0.2 |
| HLA-B*35:01-S$_{896-904}$ | IPFAMQMAY | 61.6 $\pm$ 0.6 |

[1] Tm is determined at 50% of its normalised fluorescence intensity and is indicative of the temperature required to unfold 50% of the protein. Tm values are represented as the mean $\pm$ S.E.M. of n = 2.

### 2.2. Differential Scanning Fluorimetry

Differential scanning fluorimetry (DSF) was performed to determine the stability of each pHLA using the fluorescent dye SYPRO orange, and fluorescence was measured in a Qiagen RG6 real-time PCR machine. Each of the pHLA complexes was in a solution of 10 mM Tris-HCl pH 8 and 150 mM NaCl, and was measured at two different concentrations (5 and 10 μM) in duplicate, where samples were heated from 30 to 95 $^\circ$C at a rate of 0.5 $^\circ$C/min. Fluorescence intensity was detected using a default excitation and emission channel set to yellow (excitation of approximately 530 nm and detection at approximately 557 nm). Fluorescence intensity data was normalised and plotted using GraphPad Prism 8 (version 8.4.2). The Tm, or thermal midpoint, represents the temperature at 50% of maximal fluorescence. The results are reported in Table 1.

## 2.3. Peptide Conservation within SARS-CoV-2 Isolates

Complete spike protein sequences from SARS-CoV-2 isolates (taxid ID 2697049) were obtained from the NCBI virus database http://www.ncbi.nlm.nih.gov/labs/virus (accessed on 22 March 2021). Sequences were aligned using https://www.fludb.org/brc/home.spg?decorator=influenza (accessed on 22 March 2021). Sequences with an unknown amino acid (X) within the peptide of interest were removed from the analysis. The sequence alignment results are summarised in Table 2.

**Table 2.** Sequence conservation of the three spike-derived peptides study from SARS-CoV-2 isolates.

| Peptide | Africa | Asia | Europe | North America | Oceania | South America |
|---------|--------|------|--------|---------------|---------|---------------|
| $S_{386-395}$ KLNDLCFTNV | 100% (770) | 100% (2195) | 99.89% (888) | 99.99% (49,376) | 100% (9919) | 100% (449) |
| $S_{370-378}$ NSASFSTFK | 99.87% (773) | 100% (2198) | 99.78% (889) | 99.89% (49,369) | 100% (9919) | 100% (441) |
| $S_{896-904}$ IPFAMQMAY | 100% (774) | 99.37% (2208) | 100% (889) | 99.93% (49,386) | 99.90% (9919) | 99.78% (451) |

Complete full-length sequences were obtained from the NCBI virus database http://www.ncbi.nlm.nih.gov/labs/virus (accessed on 22 March 2021) and were aligned using https://www.fludb.org/brc/home.spg?decorator=influenza (accessed on 22 March 2021). The frequency of peptide conservation is shown, with the total number of sequences aligned being: 63,597 for $S_{386-395}$, 63,589 for $S_{370-378}$, and 63,627 for $S_{896-904}$. The number of sequences from each geographic region is shown in parenthesis.

## 2.4. Crystallisation and Structural Determination

Crystals of pHLA complexes were grown via the sitting-drop vapour-diffusion method at 20 °C with a protein:reservoir drop ratio of 1:1, at a concentration range of 3.5–7 mg/mL in 10 mM Tris-HCl pH 8.0, 150 mM NaCl. Crystals of HLA-A*02:01 in complex with SARS-CoV-2 $S_{386-395}$ (KLNDLCFTNV) were grown in 14% ($w/v$) PEG 3350, 0.2 M ammonium tartrate, and 1 mM cadmium chloride. HLA-A*11:01 in complex with SARS-CoV-2 $S_{370-378}$ (NSASFSTFK) were grown in 2 M ammonium sulfate, 0.1 M sodium cacodylate pH 6.5, and 0.2 M sodium chloride. HLA-B*35:01 in complex with SARS-CoV-2 $S_{896-904}$ (IPFAMQMAY) were grown in 18% ($w/v$) PEG 3350 and 0.2 M sodium fluoride. These crystals were soaked in a cryoprotectant containing the respective mother liquor and 20% ($v/v$) ethylene glycol or 30% ($w/v$) PEG 3350 and flash-frozen in liquid nitrogen. The datasets were collected on the MX2 beamline at the Australian Synchrotron (Clayton, Australia) [11]. The data were processed using XDS [12], and the structures were solved using molecular replacement using the PHASER program [13] from the CCP4 suite [14] with models of HLA-A*02:01 [7], HLA-A*11:01 [15], and HLA-B*35:01 [16] without the peptide. Manual model building was conducted using COOT [17] and refinement was performed with BUSTER [18]. The final model was validated using the wwPDB OneDep system with the accession number of 7M8S for HLA-A*02:01 SARS-CoV-2 $S_{386-395}$, 7M8T for HLA-A*11:01 SARS-CoV-2 $S_{370-378}$, and 7M8U for HLA-B*35:01 SARS-CoV-2 $S_{896-904}$. The final refinement statistics are summarized in Table 3. All molecular graphic representations were created using PyMOL.

**Table 3.** Data collection and refinement statistics.

| Data Collection Statistics | HLA-A*02:01-S$_{386-395}$ | HLA-A*11:01-S$_{370-378}$ | HLA-B*35:01-S$_{896-904}$ |
|---|---|---|---|
| Space group | P2$_1$2$_1$2$_1$ | P2$_1$ | P2$_1$2$_1$2$_1$ |
| Cell dimensions (a,b,c) (Å) | 64.60, 86.69, 163.21 | 49.87, 38.45, 110.06 β = 94.63° | 51.24, 82.38, 111.01 |
| Resolution (Å) | 46.08–2.35 (2.43–2.35) | 46.72–1.50 (1.53–1.50) | 46.52–1.44 (1.47–1.44) |
| Total number of observations | 266,968 (25,197) | 458,019 (22,980) | 1,114,777 (47,083) |
| Number of unique observations | 39,044 (3769) | 67,146 (3290) | 84,730 (3897) |
| Multiplicity | 6.8 (6.7) | 6.8 (7.0) | 1.2 (12.1) |
| Data completeness (%) | 100 (100) | 100 (100) | 99.7 (93.6) |
| I/σ$_I$ | 10.0 (2.0) | 13.8 (1.8) | 16.4 (1.8) |
| R$_{pim}$ [a] (%) | 5.2 (40.9) | 2.9 (43.2) | 1.9 (40.5) |
| CC$_{1/2}$ (%) | 98.2 (67.0) | 99.9 (75.4) | 99.9 (76.4) |
| Refinement Statistics | | | |
| R$_{factor}$ [b] (%) | 18.1 | 18.1 | 20.4 |
| R$_{free}$ [b] (%) | 23.0 | 21.1 | 22.9 |
| rmsd from ideality | | | |
| Bond lengths (Å) | 0.010 | 0.010 | 0.010 |
| Bond angles (°) | 0.127 | 0.103 | 0.103 |
| Ramachandran plot (%) | | | |
| Favoured | 96.0 | 99.0 | 99.0 |
| Allowed | 4.0 | 1.0 | 1.0 |
| Disallowed | 0.0 | 0.0 | 0.0 |
| PBD code | 7M8S | 7M8T | 7M8U |

[a] $R_{p.i.m} = \Sigma_{hkl} [1/(N-1)]^{1/2} \Sigma_i \mid I_{hkl, i} - <I_{hkl}> \mid /\Sigma_{hkl} <I_{hkl}>$. [b] $R_{factor} = \Sigma_{hkl} \mid \mid F_o \mid - \mid F_c \mid \mid /\Sigma_{hkl} \mid F_o \mid$ for all data except approximately 5% which were used for R$_{free}$ calculation. Values in parentheses are for the highest-resolution shell.

## 3. Results

### 3.1. The Spike-Derived Peptides Were Able to Form a Stable Complex with Their Respective HLA Molecules

We selected three peptides derived from the SARS-CoV-2 spike protein that were originally predicted to bind to HLA molecules (Table 1) [19–21]. Subsequently, the S$_{386-395}$ peptide was described as recognised by CD8$^+$ T cells using tetramer staining [22], and S$_{896-904}$ can activate CD8$^+$ T cells using a T cell activation assay [23]. Therefore, they are good potential targets as T cell antigens, and warrant more investigation.

These three spike-derived peptides have been conserved in the sequenced SARS-CoV-2 isolates reported so far [24]. Indeed, our sequence analysis of spike proteins sequenced from >60,000 SARS-CoV-2 isolates revealed that all three peptides were >99% conserved in all geographic locations (Table 2). Therefore, they could represent good targets for therapeutic and vaccine design.

Our first aim was to determine if each peptide was able to form a stable complex with its specific HLA molecule. To this end, we refolded each of the three peptides with its respective HLA molecule, purified the pHLA complexes, and determined their thermal stability (Table 1). The data showed that each of the pHLA complexes had a thermal midpoint temperature, or Tm, well above the human body temperature of 37 °C. Therefore, it is expected that these pHLA complexes will remain stable on the cell surface. Interestingly, in addition to binding HLA-A*11:01, the S$_{370-378}$ peptide has also been predicted to bind to the HLA-A*03:01 molecule [21], and is described as immunogenic in HLA-A68$^+$ donors [23]. As these three HLA molecules all belong to the HLA-A3 superfamily [25], it is likely that the S$_{370-378}$ peptide is able to be presented by all three HLA molecules [26].

This data confirms that the three spike-derived peptides can form stable complexes with their respective HLA molecules, and are therefore an interesting target for T cells.

### 3.2. Structure of HLA-A*02:01-S$_{386-395}$ Reveals a Flat Peptide Conformation

To understand how SARS-CoV-2 spike-derived peptides are presented to T cells, we solved the structure of each peptide in complex with their respective HLA molecule using X-ray crystallography (Table 3).

We solved the structure of S$_{386-395}$ in complex with HLA-A*02:01 at a resolution of 2.35 Å, and the electron density was clear for the peptide (Figure 1A,B). The structure shows that the S$_{386-395}$ peptide binds into the peptide-binding cleft of HLA-A*02:01 in a canonical conformation, with anchor residues at position 2 (P2-Leu) and position 10 (P10-Val) docking deep into the B and F pockets, respectively (Figure 1C). In addition, the P3-Asn, P5-Leu, and P7-Phe were also buried, acting as secondary anchors, and interact with each other.

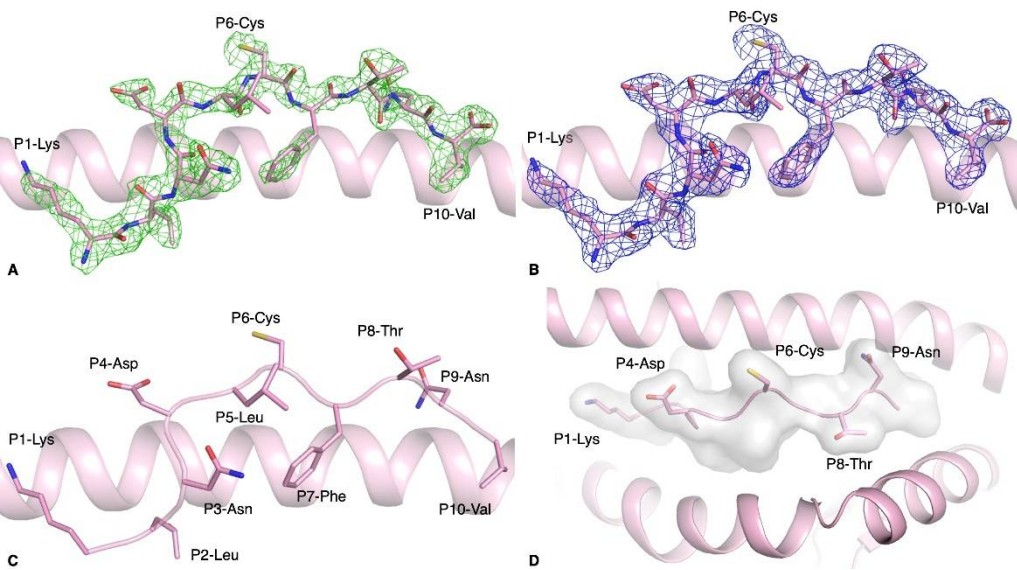

**Figure 1.** Structure of HLA-A*02:01-S$_{386-395}$ complex. (**A,B**) Electron density map of (**A**) Fo-Fc at 3σ (green) and (**B**) 2Fo-Fc at 1 σ (blue) for S$_{386-395}$ peptide (pink stick) presented by HLA-A*02:01 molecule (pink cartoon). (**C**) Side view of the S$_{386-395}$ peptide (pink stick) showing the solvent exposed and anchor residues. (**D**) Top-down view of the antigen-binding cleft of HLA-A*02:01 (pink cartoon) with the S$_{386-395}$ peptide represented as pink sticks. The white surface indicates the surface that is available for potential contact with T cells.

The S$_{386-395}$ peptide is a 10mer peptide, longer than the classical 9mer that is highly characteristic of HLA class I molecules. While 9mer peptides fit perfectly in an extended conformation in the antigen-binding cleft of HLA class I, longer peptides have adopted conformations where some residues of the peptide are bulged out of the cleft [27]. The 10mer peptide S$_{386-395}$ is not bulged out of the HLA-A*02:01 antigen-binding cleft, but adopts a rather flat conformation, likely due to the fact that half of its residues are buried in the cleft. This peptide was predicted to bind HLA-A*02:01 [19], and was recognised by T cells from unexposed donors [22]. The molecular docking prediction from Can et al. showed that P1-Lys, P3-Asn, P4-Asp, and P6-Cys were predicted to be solvent exposed. Comparison with our crystal structure revealed that the solvent-exposed residues were P1-Lys, P4-Asp, P6-Cys, P8-Thr, and P9-Asn instead (Figure 1D). Interestingly, the P6-Cys was solvent exposed and therefore available to form a disulfide bond. Indeed, we observed a disulfide bond between the peptides of two pHLA complexes contained in the crystal asymmetric unit.

### 3.3. HLA-A*11:01-S$_{370-378}$ Presents Aromatic Residues to T Cells

HLA molecules are classified into different superfamilies based on the binding properties on their peptide-binding groove. The HLA-A3 superfamily favours a small aliphatic amino acid on position 2 (P2) and a positively charged amino acid on the C-terminus of the

peptide (PΩ), which are the primary anchors of the binding groove. There are three main members in the HLA-A3 superfamily [25]: HLA-A*03:01, HLA-A*11:01, and HLA-A*68:01. The $S_{370-378}$ peptide has been predicted to bind HLA-A*03:01 and HLA-A*11:01 [21], and can activate CD8$^+$ T cells in an HLA-A*68:01 donor [23]. It has previously been reported that certain peptides can be presented by multiple HLA molecules [26], and that they can also be immunogenic [28]. Interestingly, immunogenicity of the $S_{370-378}$ peptide is still debatable, and might depend on HLA-restriction, as it has been predicted to be both immunogenic [21] and non-immunogenic [29]. Further immunogenicity studies need to be undertaken to determine if this peptide presented by HLA-A*03:01 can activate CD8$^+$ T cells. However, this peptide has been described as non-immunogenic in a small cohort of healthy and COVID-19-recovered individuals expressing HLA-A*11:01 [30], whereas it is immunogenic when presented by HLA-A*68:01 [23]. It is therefore possible that $S_{370-378}$ is either not immunogenic in HLA-A*11:01$^+$ donors alone, or is non-immunogenic in only some HLA-A*11:01$^+$ donors, and more investigation will be required to determine which of these possibilities is the case.

To gain a further understanding of how the $S_{370-378}$ peptide might be seen by CD8$^+$ T cells, we solved the structure of the HLA-A*11:01 molecule presenting this peptide (Table 3). The structure of the HLA-A*11:01-$S_{370-378}$ complex was solved at high resolution (1.50 Å), and the electron density was clear for the peptide (Figure 2A,B).

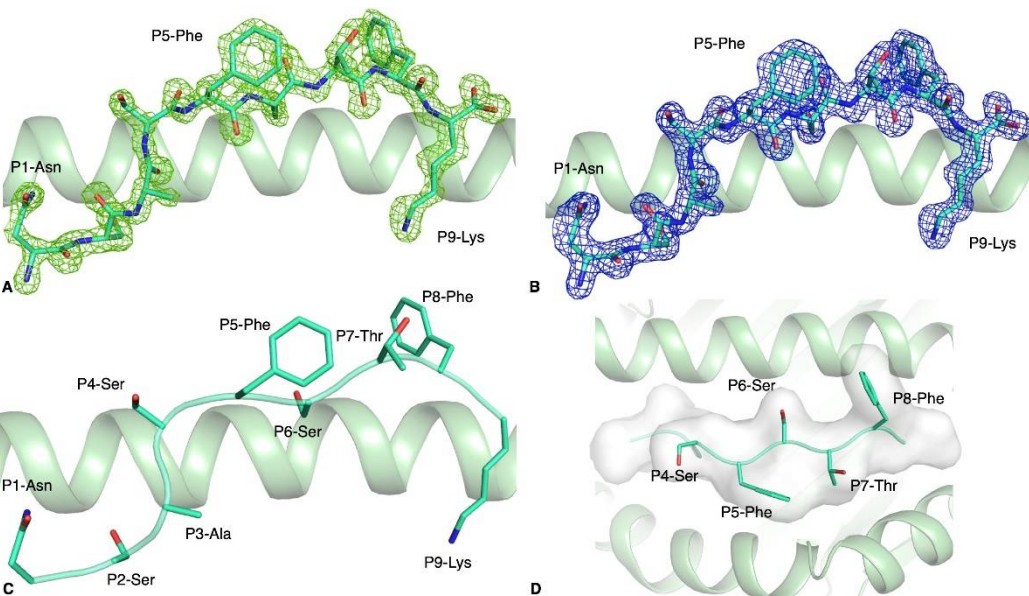

**Figure 2.** Structure of the HLA-A*11:01-$S_{370-378}$ complex. Electron density map of (**A**) Fo-Fc at 3σ (green) and (**B**) 2Fo-Fc at 1σ (blue) around the $S_{370-378}$ peptide (green stick) presented by the HLA-A*11:01 molecule (green cartoon). (**C**) Side view of the $S_{370-378}$ peptide (green stick) showing the solvent exposed and anchor residues. (**D**) Top view of the antigen-binding cleft of HLA-A*11:01 (green cartoon) with the $S_{370-378}$ peptide represented as green stick and cartoon as well as a white surface to show that the surface is available for potential contact with T cells.

The $S_{370-378}$ peptide adopted a canonical extended conformation in the cleft of the HLA-A*11:01 molecule, with P2-Ser and P9-Lys acting as secondary anchor residues (Figure 2C) without additional secondary anchor residues. The backbone of the peptide's central part (P4–P8) is flat and solvent exposed in the cleft of HLA-A*11:01. The surface exposed to the solvent (Figure 2D), and therefore available for potential T cell receptor contact, is hydrophobic with three small side chains (P4-Ser, P6-Ser, and P7-Thr) and two large aromatic residues (P5-Phe and P8-Phe). The $S_{370-378}$ peptide presents a lot of exposed residues that could be contacted by TCRs.

### 3.4. The $S_{896-904}$ Peptide Adopted a Flat Conformation in the Cleft of HLA-B*35:01

The $S_{896-904}$ peptide was also predicted to bind several HLA molecules by Al-Khafaji et al. [20]. The strongest $IC_{50}$ predicted was for HLA-B*35:01, for which the primary anchor residues of the $S_{896-904}$ peptide would be favourable (P2-Pro and P9-Tyr, Table 1). In addition, the $S_{896-904}$ peptide has been described as immunogenic in both HLA-B*51:01[+] and HLA-B*35:01[+] COVID-19-recovered donors [23]. In line with these studies, the Tm value of the HLA-B*35:01-$S_{896-904}$ complex showed a stable complex (Table 1). We solved, at high resolution (1.44 Å), the structure of the $S_{896-904}$ peptide presented by the HLA-B*35:01 molecule (Table 3), with clear electron density showing a stable conformation of the peptide (Figure 3A,B).

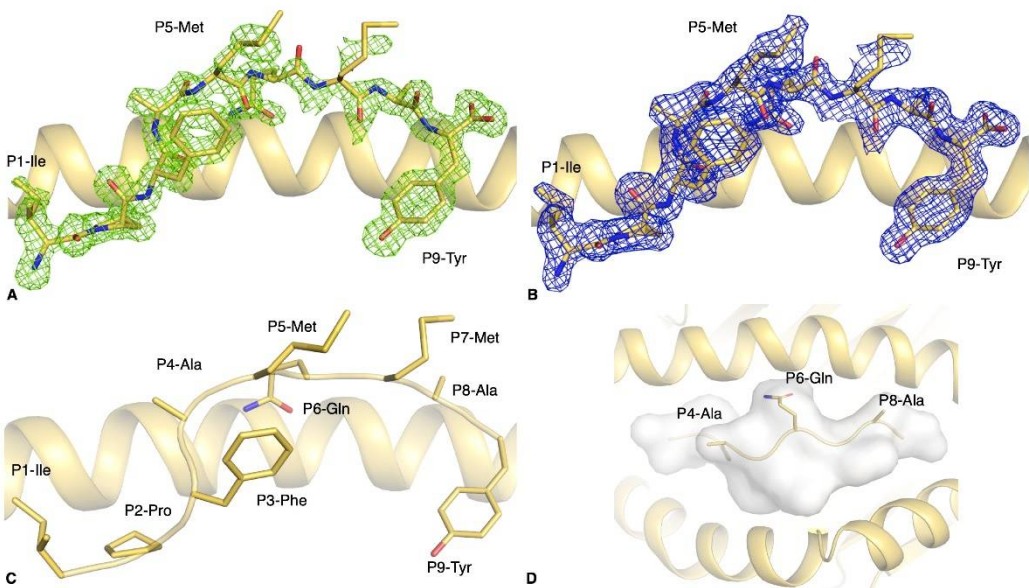

**Figure 3.** Structure of the HLA-B*35:01-$S_{896-904}$ complex. Electron density map of (**A**) Fo-Fc at 3σ (green) and (**B**) 2Fo-Fc at 1σ (blue) around the $S_{896-904}$ peptide (yellow stick) presented by the HLA-B*35:01 molecule (yellow cartoon). (**C**) Side view of the $S_{896-904}$ peptide (yellow stick) showing the solvent exposed and anchor residues. (**D**) Top view of the antigen-binding cleft of HLA-B*35:01 (yellow cartoon) with the $S_{896-904}$ peptide represented as yellow stick and cartoon, as well as a white surface to show that the surface is available for potential contact with T cells.

As predicted, the P2-Pro binds to the HLA B pocket and the P9-Tyr to the HLA F pocket, both acting as primary anchor residues, with the addition of the P3-Phe that acts as a secondary anchor (Figure 3C). Despite the presence of residues with large side chains in the central region of the peptide—namely P5-Met, P6-Gln, and P7-Met—the central part of the peptide was relatively flat in the antigen-binding cleft (Figure 3D). The two methionines at positions 5 and 7 of the peptide were half buried against the α2-helix of the HLA, while the P6-Gln buried its amide group between the peptide backbone and the α1-helix to form hydrogen bonds with the Asn70 and Thr73 of the HLA-B*35:01 molecule. Although the residues at positions 4 and 8 are solvent exposed, they are alanines and therefore only expose a methyl group, limiting its potential contact with TCRs.

## 4. Discussion

The immune response to SARS-CoV-2 infection is still an intense area of research and requires a better understanding of the differences in disease progression between individuals, as well as better identification of immunogenic antigens that can provide protective immunity. CD8[+] T cells have a critical role in viral infection, and while their part in COVID-19 is not fully understood, they are able to recognise epitopes from SARS-CoV-2 and play a role in the overall immune response [3,4,8,10,30–36]. HLA molecules are the targets of CD8[+] T cells as they present viral peptides to signal infection. As T cells recognize

a peptide bound to an HLA molecule, it is important to understand which peptides from SARS-CoV-2 will be presented to T cells, as well as their HLA restriction. Since HLA molecules are extremely polymorphic, we hereby report the analysis of three frequently expressed HLA alleles within the population.

Here, we confirmed the restriction of three spike-derived peptides to their predicted HLA molecules by refolding each HLA with a peptide and assessing the overall stability of each pHLA complex formed. All three pHLA complexes were stable and had a thermal midpoint well over physiological body temperature, suggesting that these pHLAs can remain stable on the cell surface, and would have the potential to be contacted by TCRs. In addition, we solved the crystal structure of these three pHLA complexes, showing how each peptide is presented by its specific HLA molecule. This information is important as the spike protein is prone to mutations [24,37–39]. The peptides under investigation in our study are not located within the region of spike that is mutated in the new variants such as the ones from the UK (B1.1.7), South Africa (B1.1.3), or Brazil (P1). However, as more mutations are likely to arise, it is important to understand which mutations could represent an escape from the immune system or from the currently available vaccines. For example, mutation of the residue located at the second or last position of the peptide could have a devastating impact on the ability of a peptide to bind a designated HLA molecule, which would lead to viral escape due to the lack of presentation. Residues that are solvent exposed could instead directly impact T cell recognition. Therefore, we could predict the impact a mutation might have on T cell recognition, and anticipate its effects on the immune response. In addition, the spike-derived peptides studied here are able to be presented by multiple HLA molecules, which in turn could be an advantage at a population level as some HLA could be able to bind certain variants while others could not. The structure of each peptide reveals which residue might be important for T cell recognition, which could in turn provide information about the mutations within the spike protein that might impact on T cell binding, HLA binding, and whether they are likely to escape T cell surveillance.

Altogether, our work provides insight into the spike protein-derived SARS-CoV-2 peptide presentation by HLA molecules, which could help provide a better understanding of the T cell response to the virus.

**Author Contributions:** Conceptualization, S.G.; methodology and investigation, A.T.N., C.S., D.J., C.A.L., H.H., D.S.M.C., E.J.G., and S.G.; validation, A.T.N., C.S., D.S.M.C., E.J.G. and, S.G.; formal analysis, A.T.N., E.J.G., and S.G.; resources, S.G.; data curation, E.J.G.; writing—original draft preparation, A.T.N. and S.G.; writing—review and editing, A.T.N., C.S., D.S.M.C., E.J.G. and S.G.; visualization, A.T.N. and S.G.; supervision, C.S., D.S.M.C., E.J.G. and S.G.; project administration, S.G.; funding acquisition, A.T.N., C.S., D.S.M.C., E.J.G. and S.G. All authors have read and agreed to the published version of the manuscript.

**Funding:** This work was supported by financial contributions from Monash University, Australian Nuclear Science and Technology Organisation (ANSTO, AINSE ECR grants), Australian Research Council (ARC), National Health and Medical Research Council (NHMRC), and the Medical Research Future Fund (MRFF). E.J.G. is supported by an Australian Research Council DECRA (DE210101479), S.G. is supported by an NHMRC SRF (#1159272).

**Data Availability Statement:** The final crystal structure models for the HLA-A*02:01 complexes have been deposited to the Protein Data Bank (PDB) under the following accession codes: 7M8S for HLA-A*02:01 SARS-CoV-2 $S_{386-395}$, 7M8T for HLA-A*11:01 SARS-CoV-2 $S_{370-378}$, and 7M8U for HLA-B*35:01 SARS-CoV-2 $S_{896-904}$.

**Acknowledgments:** The authors would like to thank the Monash Macromolecular Crystallisation Facility and the MX team for assistance at the Australian Synchrotron.

**Conflicts of Interest:** The authors declare no conflict of interest.

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
