# Peer review of "SARS-CoV-2 Spike-Derived Peptides Presented by HLA Molecules"

_biophysica, doi:10.3390/biophysica1020015_

Round 1

Reviewer 1 Report

In this study, Nguyen A.T. et present structural data of HLA-A*02:01, HLA-11:01 and B*35:01 complexed with specific SARS-CoV-2 peptides.   The SARS-CoV-2 peptides chosen where based on previous studies characterizing SARS-CoV-2 peptide-binding characterized by in silico approaches and in vitro recall assays.  Although theoretical computer  programs provide investigative “leads” as to what peptides may bind and be presented by HLA, these results need to be confirmed experimentally.   In this study, the authors provide/add to the body of work related to HLA-peptide crystal structures providing important 3-D data as to what SARS-CoV-2 peptides bind certain HLA molecules.  In addition, based on a select number of structural studies, the grouping of HLA supertypes could be argumentative when determining how small differences may impact which peptides are presented by certain HLA subtypes.  Regardless, these results provide important information as to which viral components may induce a clinically effective response in the context of specific HLA epitopes, which in turn should provide significant insights into human T cell responses to this pathogen.

Comment

The authors should note where and how they derived the sequences for HLA-A*02:01, HLA-A*11:01, and HLA-B*35:01.  Was this confirmed and by what methodology?

Author Response

We thank the reviewer for their positive comment, and have address the question.

The authors should note where and how they derived the sequences for HLA-A*02:01, HLA-A*11:01, and HLA-B*35:01.  Was this confirmed and by what methodology?

The sequences for HLA-A*02:01, HLA-A*11:01, and HLA-B*35:01 were obtained from the IMGT/HLA database (https://www.ebi.ac.uk/ipd/imgt/hla/), using only the soluble part of the HLA heavy chains for each (1-275 residues). Each constructs were ordered from genscript sub-cloned into pET30 vector for bacterial expression using NdeI/HindIII restriction enzyme site for the sub-cloning. Genscript provide a full sequencing of each insert after sub-cloning, confirming the sequences for each constructs. This has now been included in the materials and methods.

Reviewer 2 Report

Well done!

No further improvements. 

Author Response

We thank the reviewer for their positive message, and there was no comment to address.